# Half-Wave Potentials and In Vitro Cytotoxic Evaluation of 3-Acylated 2,5-*Bis*(phenylamino)-1,4-benzoquinones on Cancer Cells

**DOI:** 10.3390/molecules24091780

**Published:** 2019-05-08

**Authors:** Julio Benites, Jaime A. Valderrama, Maryan Ramos, Maudy Valenzuela, Angélica Guerrero-Castilla, Giulio G. Muccioli, Pedro Buc Calderon

**Affiliations:** 1Química y Farmacia, Facultad de Ciencias de la Salud, Universidad Arturo Prat, Casilla 121, Iquique 1100000, Chile; jaimeadolfov@gmail.com (J.A.V.); maryan.ramos01@gmail.com (M.R.); maudyta@gmail.com (M.V.); anguerrero@unap.cl (A.G.-C.); pedro.buccalderon@uclouvain.be (P.B.C.); 2Instituto de Ciencias Exactas y Naturales, Universidad Arturo Prat, Casilla 121, Iquique 1100000, Chile; 3Bioanalysis and Pharmacology of Bioactive Lipids (BPBL), Louvain Drug Research Institute, Université catholique de Louvain, 72 Avenue E. Mounier, BPBL 7201, 1200 Brussels, Belgium; giulio.muccioli@uclouvain.be; 4Research Group in Metabolism and Nutrition, Louvain Drug Research Institute, Université catholique de Louvain, 73 Avenue E. Mounier, 1200 Brussels, Belgium

**Keywords:** acylated 2,5-*bis*(phenylamino)-1,4-benzoquinones, oxidative amination, half-wave potential, cancer cells, cytotoxicity

## Abstract

A broad range of 3-acyl-2,5-*bis*(phenylamino)-1,4-benzoquinones were synthesized and their voltammetric values, as well as in vitro cancer cell cytotoxicities, were assessed. The members of this series were prepared from acylbenzoquinones and phenylamines, in moderate to good yields (47–74%), through a procedure involving a sequence of two in situ regioselective oxidative amination reactions. The cyclic voltammograms of the aminoquinones exhibit two one-electron reduction waves to the corresponding radical-anion and dianion, and two quasi-reversible oxidation peaks. The first and second half-wave potential values (E_1/2_) of the members of the series were sensitive to the push-pull electronic effects of the substituents around the benzoquinone nucleus. The in vitro cytotoxic activities of the 3-acyl-2,5-*bis*(phenylamino)-1,4-benzoquinones against human cancer cells (bladder and prostate) and non-tumor human embryonic kidney cells were measured using the MTT colorimetric method. The substitution of both aniline groups, by either methoxy (electron donating effect) or fluorine (electron withdrawal effect), decreased the cytotoxicity in the aminoquinones. Among the members of the unsubstituted phenylamino series, two of the 18 compounds showed interesting anti-cancer activities. A preliminary assay, looking for changes in the expression of selected genes, was performed. In this context, the two compounds increased TNF gene expression, suggesting an association with an inflammatory-like response.

## 1. Introduction

The molecular frameworks of diverse naturally-occurring cytotoxic compounds, such as smenospongine [1], streptonigrin [2], and mansouramicyn C [3], contain the aminoquinoid moiety as their key structural component (Figure 1). In the field of synthetic cytotoxic aminoquinones, those derived from 1,4-naphthoquinone [4,5,6,7,8] and their heterocyclic analogues [9,10,11,12,13,14,15,16,17] have received considerable attention. However, the synthesis and cytotoxic evaluation of aminobenzoquinones has hitherto received little attention and was mainly focused on the 2,5-*bis*(arylamino)-1,4-benzoquinone derivatives as compounds **A** and **B**. The earlier findings of Verter and Rogers [18] reported that specific 2,5-*bis*(alkylamino)-3,6-dimethoxy-1,4-benzoquinones, prepared from polyporic acid, increased the survival of L1210-bearing mice by 50%. In addition, Sarcoma 180 accumulated slower in treated Swiss albino mice than in control animals [19]. Further studies, described by Mathew et al. [20], on 2,5-*bis*(phenylamino)benzoquinone **A** and the corresponding chlorine analogue **B**, demonstrated a good in vitro inhibitory activity against human colon adenocarcinoma proliferation. Compound **B** was also found to be active against the leukemia L1210 screening in vitro.

It is worth noting that synthetic 2,5-*bis*(arylamino)-1,4-benzoquinones containing acyl and alkoxycarbonyl substituents at C-3 have been used as precursor of benz- and naphthoisoxazolequinones [21,22,23]. Some of the members of these heterocycle series exhibit in vitro activity as radiosensitizers and cytotoxic properties. Here, we wish to report the synthesis of a broad variety of 3-acyl-2,5-*bis*(arylamino)-1,4-benzoquinones, in order to give information on their redox properties and in vitro cytotoxic activity on cancer cells.

## 2. Results and Discussion

The members of the series of 3-acyl-2,5-*bis*(phenylamino)-1,4-benzoquinones (Scheme 1) were prepared from 1,4-benzoquinone and a set of aldehydes and phenylamines. The acylhydroquinone precursors **1a**–**q** were synthesized by the solar Friedel Crafts photoacylation of quinone **1** with aldehydes, according to our previously-reported procedure [24]. The preparation of 3-acyl-2,5-*bis* (phenylamino)-1,4-benzoquinones **2a**–**q** was accomplished through a sequence involving: (a) Oxidation of the acylhydroquinones **1a**–**q**, to the respective acylquinones, with silver (I) oxide (2.5 equiv.) in dichloromethane (DCM) and (b) oxidative amination of the 2-acylquinones resulting in (b) with phenylamines (2 equiv.) in ethanol under aerobic conditions. This procedure provides the substituted benzoquinones **2a**–**q** in the yield range 47–74% (Table 1), and is a modification of the procedure, described by Shäfer and Aguado in [25], for the synthesis of compound **2a** and some of their substituted phenylamino analogs.

The structures of compounds **2a**–**q** were established by ^1^H- and ^13^C-nuclear magnetic resonance (NMR), bi-dimensional nuclear magnetic resonance (2D-NMR), and high resolution mass spectroscopy (HRMS). Heteronuclear multiple bond correlation (HMBC) experiments of the members of the series, as those of the representative congeners **2b** and **2p** (Figure 2), allowed us to establish the location of the substituents around the benzoquinone nuclei. 

It is worth mentioning that inspection of minimal energy conformation of compounds **2b** and **2p**, performed by MM2 calculation (ChemBio3D 11.0, PerkinElmer, MA, USA), shows a co-planar orientation between the benzoquinone nucleus and the phenylamino substituent at C-5 (Appendix A). Furthermore, it was also observed that rotation of the substituents linked to the 2,3-quinone double bond are strongly hindered (Figure 3).

The members of the 3-acyl-2,5-*bis*(phenylamino)-1,4-benzoquinone series **2a**–**q** were evaluated for their half-wave potentials (E^I^
_1/2_ and E^II^
_1/2_). They were measured by cyclic voltammetry in acetonitrile at room temperature, using a platinum electrode and 0.1 M tetraethylamoniumtetrafluoroborate as the supporting electrolyte [26]. The voltammograms were run in the potential range from 0.0 to −2.0 V versus non-aqueous Ag/Ag^+^. The cathodic peaks related to the reduction of quinone, and the anodic one due to its re-oxidation, were observed for the compounds as well-defined quasi-reversible waves. 

The E^I^
_1/2_ values for the first one-electron, which is related with the formation of the semiquinone radical anion, fell within the range of −850 to −500 mV. The E^I^
_1/2_ values for the second one-electron transfer, corresponding to the dianion formation [27,28], were located within the range of −1170 to −860 mV (Table 1). Taking into account the notable differences of the E_1/2_ values for each of the one-electron transfer processes, it is evident that they are due to the push-pull electronic effects of the substituents located in the 1,4-benzoquinone nucleus. Table 1, regarding the E^I^
_1/2_ and E^II^
_1/2_ values of the unsubstituted phenylamino members **2a**–**h**, indicates that the nature of the acyl substituents mainly affects the second half-wave potential (ΔE^II^
_1/2_ = 300 mV) over the first one (ΔE^II^
_1/2_ = 90 mV). The effect of the insertion of methoxy substituents into the 3-acyl-2,5-*bis*(phenylamino)-1,4-benzoquinone scaffold, as in compounds **2i**–**2m**, was related to a significant cathodic shift of the E^I^
_1/2_ when compared to E^II^
_1/2_. This fact could be attributed to the donor effect of the 3,4,5-trimethoxyphenyl group, which is transmitted to the quinonoid nucleus through the amino group.

The effects on redox properties and lipophilia by either electron-donating or electron-withdrawing groups in the 3-acyl-2,5-*bis*(phenylamino)-1,4-benzoquinone scaffold (compounds **2i**–**2m** versus **2n**–**2q**) were also examined. Table 1 shows that the methoxy group (as in compounds **2i**–**2m**) led to increased values of E^I^
_1/2_ (−610 to −850 mV) but reduced values of ClogP (−0.2 to 1.19). Conversely, the fluorine group (as in compounds **2n**–**2q**) led to decreased values of E^I^
_1/2_ (−550 to −730 mV) but enhanced values of ClogP (0.88 to 2.26). As compared to previous series, it should be noted that the E^I^
_1/2_ values of the unsubstituted phenylamino members **2a**–**h** ranged between −500 and −590 mV.

The 3-acyl-2,5-*bis*(phenylamino)-1,4-benzoquinones **2a**–**q** were evaluated for their in vitro cytotoxic activities against normal human embryonic kidney cells (HEK-293 cells) and two human cancer cell lines (T24 and DU-145 cells) in 72 h drug exposure assays. The cytotoxic activities of the new compounds were measured using conventional microculture tetrazolium reduction assays [29]. The cytotoxic activities are expressed in terms of IC_50_. Doxorubicin, a clinically used anti-cancer agent, was taken as a positive control. The cytotoxic activity data are summarized in Table 2.

Regarding the cytotoxicity of aminoquinones, it should be noted that the DU-145 cells were more sensitive than T24 and HEK-293 cells. When comparing the IC_50_ values calculated for the most active aminoquinones in both cancer cell lines, their activity ranged from 16.3 to 51.80 µM. Such values were at least one order of magnitude higher than that obtained with doxorubicin (0.43 and 0.93 µM in T24 and DU-145 cells, respectively). The unsubstituted phenylamino members **2a**–**2h** were largely more active than the compounds **2i**–**2q**, showing that the substitution of both aniline groups—by either methoxy (electron donating effect) or fluorine (electron withdrawal effect)—decreased the cytotoxicity of the aminoquinones. Among the series **2a**–**2h**, the compounds **2c** (IC_50_ values of 16.3 and 45.2 µM in T24 and DU-145 cells, respectively) and **2d** (IC_50_ values of 34.0 and 23.5 µM in T24 and DU-145 cells, respectively) were the most active; displaying, in addition, a highly selective effect, as HEK-293 cells were affected only at doses higher than 100 µM. Moreover, within the same series, compounds **2f** and **2g** showed similar cytotoxic activities to previous compounds, but with a much lower selectivity. Outside this series, compound **2l** was also active, but without any selectivity as it affected both HEK-293 and cancer cells. Thus, it may be concluded that, among all the tested aminoquinones, congeners **2c** and **2d** displayed the best cytotoxic activities, exhibiting high selectivity and lipophilicity values. Therefore, they represent lead-molecules for further investigations in exploring both intracellular targets as well as their molecular mechanism of action.

In order to gain information about potential molecular targets, a preliminary assay for gene expression was conducted. To this end, some genes were selected, regarding their key role in cell survival, for instance mTOR, TP53, TNF, and so on. Table 3 shows the relative expression levels of the genes implicated in anti-cancer effects in T24 cells after treatment with **2c** and **2d**.

Compound **2c** (and, to a lesser extent, **2d**) enhanced the expression of the TNF gene. Prior to drawing a definitive conclusion, such an increase in gene expression should be further confirmed by measuring its protein levels and activity. Nevertheless, it should be emphasized that a local increase in TNF concentration is not only associated with an inflammatory response, but also with an immunogenic response able to activate tumor-specific cytotoxic T lymphocytes, which can seek out and destroy tumor cells and reduce tumor lesions [30]. In this context, the compounds **2c** and **2d** display a potential application in cancer immunotherapy that deserves to be further investigated. 

## 3. Materials and Methods 

### 3.1. General Information

All the solvents and reagents were purchased from different companies, such as Aldrich (St. Louis, MO, USA) and Merck (Darmstadt, Germany), and were used as supplied. Melting points (mp) were determined on a Stuart Scientific SMP3 (Staffordshire, UK) apparatus and are un-corrected. The IR spectra were recorded on an FT IR Bruker spectrophotometer, model Vector 22 (Bruker, Rheinstetten, Germany), using KBr disks, and the wave numbers are given in cm^−1^. ^1^H- and ^13^C-NMR spectra were recorded on a Bruker Avance-400 instrument (Bruker, Ettlingen, Germany) in CDCl_3_ or DMSO-*d*_6_ at 400 and 100 MHz, respectively. Chemical shifts are expressed in ppm downfield relative to tetramethylsilane, and the coupling constants (*J*) are reported in Hertz. Data for the ^1^H-NMR spectra are reported as follows: s = singlet, br s = broad singlet, d = doublet, m = multiplet, and the coupling constants (*J*) are in Hz. Bi-dimensional NMR techniques and distortion-less enhancement by polarisation transfer (DEPT) were used for the signal assignment. Chemical shifts are expressed in ppm downfield relative to tetramethylsilane, and the coupling constants (*J*) are reported in Hertz. The HRMS data for all final compounds were obtained using a LTQ-Orbitrap mass spectrometer (Thermo-Fisher Scientific, Waltham, MA, USA) with the analysis performed using an atmospheric-pressure chemical ionization (APCI) source, operated in positive mode. Silica gel Merck 60 (70–230 mesh, from Merck) was used for preparative column chromatography and thin layer chromatography (TLC) aluminum foil 60F_254_ was used for analytical thin layer chromatography. The acylbenzohydroquinones (**1a**–**q**) were prepared according to a previously-reported procedure [24].

### 3.2. Chemistry

#### Preparation of 3-Acyl-2,5-bis(phenylamino)-1,4-benzoquinones **2a**–**q**, General Procedure. 

Suspensions of the acylhydroquinones **1a**–**q** (1 equiv.), Ag_2_O (2.0 equiv.) and MgSO_4_ anhydrous (300 mg) in dichloromethane (30 mL) were left with stirring for 30 min at room temperature (rt). The mixtures were filtered, the solids were washed with dichloromethane (3 × 25 mL), and the filtrates were evaporated under reduced pressure. The residues were dissolved in ethanol, the phenylamines (2 equiv.) were added to the solutions, and the mixtures were left with stirring at rt for 24 h. The solvents were removed under reduced pressure and the residues were column-chromatographed over silica gel (petroleum ether/EtOAc) to yield the corresponding pure 3-acyl-2,5-*bis*(phenylamino)-1,4-benzoquinones **2a**–**q**.

*3-Acetyl-2,5-bis(phenylamino)-1,4-benzoquinone***2a** (64%); red solid; mp: 200–201 °C. ^1^H-NMR (DMSO-*d*_6_): δ 2.30 (s, 1H, Me), 5.80 (s, 1H, quinone), 7.22 (m, 4H, arom.), 7.36 (m, 4H, arom.), 7.45 (dd, 2H, *J* = 7.5, 7.7 Hz, arom.), 9.55 (s, 1H, 5-N**H**Ph), 11.31 (br s, 1H, 2-N**H**Ph). ^13^C-NMR (DMSO-*d*_6_): δ 31.9, 96.4, 109.1, 123.9 (3C), 124.2 (2C), 125.8, 126.2, 128.7 (2C), 129.3 (2C), 137.6, 139.2, 147.6, 178.3, 178.5, 199.3. HRMS (APCI): [M + H]^+^ calcd. for C_20_H_16_N_2_O_3_: 332.11609; found 332.11359.

*3-Butiryl-2,5-bis(phenylamino)-1,4-benzoquinone***2b** (56%); red solid; mp: 115–116 °C. ^1^H-NMR (DMSO-*d*_6_): δ 0.78 (t, 3H, *J* = 7.4 Hz, -CH_2_-CH_2_-**CH_3_**), 1.23 (m, 2H, -CH_2_-**CH_2_**-CH_3_), 2.63 (t, 2H, *J* = 7.4 Hz, -**CH_2_**-CH_2_-CH_3_), 5.81 (s, 1H, quinone), 7.15 (d, 2H, *J* = 7.4 Hz, H-arom.), 7.20 (t, 1H, *J* = 7.4 Hz, H-arom.), 7.25 (t, 1H, *J* = 7.3 Hz, H-arom.), 7.32 (t, 2H, *J* = 7.8 Hz, H-arom.), 7.38 (d, 2H, *J* = 7.4 Hz, H-arom.), 7.45 (t, 2H, *J* = 7.8 Hz, H-arom.), 9.51 (s, 1H, 5-N**H**Ph), 10.85 (br s, 1H, 2-N**H**Ph). ^13^C-NMR (DMSO-*d*_6_): δ 13.7, 16.6, 45.6, 96.0, 109.2, 123.9 (2C), 124.4 (2C), 125.8, 126.2, 128.7 (2C), 129.3 (3C), 137.6, 139.0, 147.5, 178.2, 178.8, 201.6. HRMS (APCI): [M + H]^+^ calcd. for C_22_H_20_N_2_O_3_: 360.14739; found 360.15469.

*3-Hexanoyl-2,5-bis(phenylamino)-1,4-benzoquinone***2c** (57%); brown solid; mp: 140–141 °C. ^1^H-NMR (DMSO-*d*_6_): δ 0.83 (t, 3H, *J* = 7.1 Hz, -CH_2_-CH_2_-CH_2_-CH_2_-**CH_3_**), 1.17 (m, 6H, -CH_2_-**CH_2_-CH_2_-CH_2_**-CH_3_), 2.63 (t, 2H, *J* = 7.0 Hz, -**CH_2_**-CH_2_-CH_2_-CH_2_-CH_3_), 5.80 (s, 1H, quinone), 7.14 (d, 2H, *J* = 7.4 Hz, H-arom.), 7.20 (t, 1H, *J* = 7.4 Hz, H-arom.), 7.25 (t, 1H, *J* = 7.3 Hz, H-arom.), 7.32 (t, 2H, *J* = 7.7 Hz, H-arom.), 7.38 (d, 2H, *J* = 7.4 Hz, H-arom.), 7.45 (t, 2H, *J* = 7.8 Hz, H-arom.), 9.51 (s, 1H, 5-N**H**Ph), 10.79 (br s, 1H, 2-N**H**Ph). ^13^C-NMR (DMSO-*d*_6_): δ 13.8, 22.0, 22.8, 30.8, 43.5, 95.9, 109.3, 123.9 (2C), 124.4 (2C), 125.8, 126.2, 128.8 (2C), 129.3 (3C), 137.6, 139.0, 147.5, 178.2, 178.8, 201.6. HRMS (APCI): [M + H]^+^ calcd. for C_24_H_24_N_2_O_3_: 388.17869; found 388.18599.

*3-Octanoyl-2,5-bis(phenylamino)-1,4-benzoquinone***2d** (59%); brown solid; mp: 91–92 °C. ^1^H-NMR (DMSO-*d*_6_): δ 0.86 (t, 3H, *J* = 7.0 Hz, -(CH_2_)_6_C**H**_3_, 1.23 (m, 10H, -CH_2_-(C**H**_2_)_5_CH_3_), 2.63 (t, 2H, *J* = 6.5 Hz, -C**H**_2_(CH_2_)_5_CH_3_), 5.81 (s, 1H, quinone), 7.14 (d, 2H, *J* = 7.5 Hz, H-arom.), 7.19 (t, 1H, *J* = 7.4 Hz, H-arom.), 7.25 (t, 1H, *J* = 7.2 Hz, H-arom.), 7.32 (t, 2H, *J* = 7.7 Hz, H-arom.), 7.38 (d, 2H, *J* = 7.4 Hz, H-arom.), 7.45 (t, 2H, *J* = 7.8 Hz, H-arom.), 9.51 (s, 1H, 5-N**H**Ph), 10.81 (br s, 1H, 2-N**H**Ph). ^13^C-NMR (DMSO-*d*_6_): δ 13.9, 22.0, 23.1, 28.5, 28.6, 31.1, 43.6, 95.9, 109.3, 123.8 (2C), 124.4 (2C), 125.8, 126.1, 128.7 (2C), 129.3 (3C), 137.6, 139.0, 147.5, 178.2, 178.8, 201.6. HRMS (APCI): [M + H]^+^ calcd. for C_26_H_28_N_2_O_3_: 416.20999; found 416.21710.

*3-(3,4-Dimethoxybenzoyl)-2,5-bis(phenylamino)-1,4-benzoquinone***2e** (55%); gray solid; mp: 230.5–231.5 °C. ^1^H-NMR (DMSO-*d*_6_): δ 3.67 (s, 3H, OMe), 3.82 (s, 3H, OMe), 5.91 (s, 1H, quinone), 6.81 (t, 3H, *J* = 7.6 Hz, H-arom.), 6.94 (m, 4H, H-arom.), 7.26 (t, 2H, *J* = 7.2 Hz, H-arom.), 7.41 (d, 2H, *J* = 7.7 Hz, H-arom.), 7.46 (t, 2H, *J* = 7.4 Hz, H-arom.), 9.42 (s, 1H, 5-N**H**Ph), 9.54 (br s, 1H, 2-N**H**Ph). ^13^C-NMR (DMSO-*d*_6_): δ 55.3, 55.7, 95.3, 107.7, 109.9, 110.3, 123.8 (2C), 124.0, 125.7 (2C), 125.7, 126.0, 128.0 (2C), 129.3 (3C), 130.9, 137.7, 146.0, 147.3, 148.3, 152.8, 178.4, 179.3, 191.5. HRMS (APCI): [M + H]^+^ calcd. for C_27_H_22_N_2_O_5_: 454.15287; found 454.15964.

*3-(3,4,5-Trimethoxybenzoyl)-2,5-bis(phenylamino)-1,4-benzoquinone***2f** (58%); brown solid; mp: 216.5–217.5 °C. ^1^H-NMR (CDCl_3_): δ 3.80 (s, 6H, OMe), 3.89 (s, 3H, OMe), 6.19 (s, 1H, quinone), 6.74 (s, 2H, H-arom.), 6.82 (m, 2H, H-arom.), 7.06 (m, 3H, H-arom.), 7.28 (m, 3H, H-arom.), 7.44 (t, 2H, *J* = 7.8 Hz, H-arom.), 8.23 (s, 1H, 5-N**H**Ph), 8.48 (br s, 1H, 2-N**H**Ph). ^13^C-NMR (CDCl_3_): δ 56.4 (2C), 61.0, 106.5 (2C), 123.1 (3C), 125.7 (2C), 126.6, 127.1, 129.0 (2C), 129.9 (3C), 133.1, 136.7, 136.9, 142.8, 146.0, 146.7, 152.9 (2C), 178.3, 179.3, 191.8. HRMS (APCI): [M + H]^+^ calcd. for C_28_H_24_N_2_O_6_: 484.16344; found 484.16364.

*3-(Furan-2-carbonyl)-2,5-bis(phenylamino)-1,4-benzoquinone***2g** (56%); brown solid; mp: 141.5–142.5 °C. ^1^H-NMR (DMSO-*d*_6_): δ 5.89 (s, 1H, quinone), 6.88 (d, 2H, *J* = 7.4 Hz, H-arom.), 7.01 (m, 3H, H- arom), 7.09 (t, 1H, *J* = 4.3 Hz, H-arom.), 7.26 (t, 1H, *J* = 7.1 Hz, H-arom.), 7.41 (d, 2H, *J* = 7.7 Hz, H-arom.), 7.46 (t, 2H, *J* = 7.7 Hz, H-arom.), 7.58 (d, 1H, *J*= 3.5 Hz, H-arom.), 7.84 (d, 1H, *J* = 4.7 Hz, H-arom.), 9.45 (s, 1H, 5-N**H**Ph), 9.65 (br s, 1H, 2-N**H**Ph). ^13^C-NMR (DMSO-*d*_6_): δ 95.3, 107.7, 123.9 (2C), 125.6 (2C), 125.8, 126.2, 127.9, 128.0 (2C), 129.3 (2C), 133.9, 134.8, 137.5, 137.6, 145.2, 146.0, 147.4, 178.1, 179.0, 184.8. HRMS (APCI): [M + H]^+^ calcd. for C_23_H_16_N_2_O_4_: 384.11101; found 384.11164.

*3-(Thiophen-2-carbonyl)-2,5-bis(phenylamino)-1,4-benzoquinone***2h** (60%); brown solid; mp: 245–246 °C. ^1^H-NMR (DMSO-*d*_6_): δ 5.88 (s, 1H, quinone), 6.57 (d, 1H, *J* = 3.3 Hz, H-arom.), 6.90 (m, 2H, H- arom), 7.00 (d, 1H, *J* = 3.2 Hz, H-arom.), 7.04 (d, 3H, *J* = 5.1 Hz, H-arom.), 7.26 (t, 1H, *J* = 7.1 Hz, H-arom.), 7.40 (d, 2H, *J* = 7.9 Hz, H-arom.), 7.46 (t, 2H, *J* = 7.5 Hz, H-arom.), 7.81 (s, 1H, H-arom.), 9.46 (s, 1H, 5-N**H**Ph), 9.68 (br s, 1H, 2-N**H**Ph). ^13^C-NMR (DMSO-*d*_6_): δ 95.3, 107.2, 112.5, 118.2, 123.9 (2C), 125.2 (2C), 125.8, 126.3, 128.1 (2C), 129.3 (2C), 137.6, 137.7, 146.2, 146.8, 147.4, 153.1, 178.1, 179.0, 180.0. HRMS (APCI): [M + H]^+^ calcd. for C_23_H_16_N_2_O_3_S: 400.08816; found 400.09555.

*3-Benzoyl-2,5-bis-[(3,4,5-trimethoxyphenyl)amino]-1,4-benzoquinone***2i** (74%); brown solid; mp: 224–225 °C. ^1^H-NMR (CDCl_3_) δ: 3.54 (s, 3H, OMe), 3.76 (s, 3H, OMe), 3.80 (s, 3H, OMe), δ: 3.87 (s, 9H, 3 × OMe), 5.97 (s, 2H, H-arom.) 6.14 (s, 1H, quinone), 6.49 (s, 2H, H-arom.), 7.31 (m, 1H, H-arom.), 7.45 (m, 1H, H-arom.), 7.56 (d, 2H, *J* = 7.3 Hz, H-arom.), 8.16 (s, 1H, 5-N**H**Ph), 8.42 (br s, 1H, 2-N**H**Ph). ^13^C-NMR (CDCl_3_) δ: 55.8 (2C), 56.4 (2C), 60.9, 61.1, 95.9, 101.0 (2C), 102.9 (2C), 107.0, 128.4 (2C), 128.8 (2C), 132.3, 132.5, 133.0, 136.7, 136.9, 137.6, 145.9, 147.0, 153.2 (2C), 154.1 (2C), 178.2, 179.0, 192.9. HRMS (APCI): [M + H]^+^ calcd. for C_31_H_30_N_2_O_9_: 574.19513; found 574.20187.

*3-(4-Methoxybenzoyl)-2,5-bis-[(3,4,5-trimethoxyphenyl)amino]-1,4-benzoquinone***2j** (57%); red solid; mp: 190–191 °C. ^1^H-NMR (CDCl_3_) δ: 3.57 (s, 6H, 2 × OMe), 3.77 (s, 3H, OMe), 3.82 (s, 3H, OMe), 3.87 (s, 9H, 3 × OMe), 5.99 (s, 2H, H-arom.), 6.11 (s, 1H, quinone), 6.49 (s, 2H, H-arom.), 6.77 (d, 2H, *J* = 8.8 Hz, H-arom.), 7.53 (d, 2H, *J* = 8.8 Hz, H-arom.), 8.16 (s, 1H, 5-N**H**Ph), 8.37 (br s, 1H, 2-N**H**Ph). ^13^C-NMR (CDCl_3_) δ: 55.6, 55.8 (2C), 56.4 (2C), 60.8, 61.1, 95.8, 101.0 (2C), 102.9 (2C), 107.3, 113.5 (2C), 131.0, 131.2 (2C), 132.4, 132.6, 136.7, 136.8, 145.6, 147.0, 153.0 (2C), 154.0 (2C), 163.5, 178.2, 179.2, 191.3. HRMS (APCI): [M + H]^+^ calcd. for C_32_H_32_N_2_O_10_: 604.20570; found 604.21238.

*3-(4-Hydroxy-3-methoxybenzoyl)-2,5-bis-[(3,4,5-trimethoxyphenyl)amino]-1,4-benzoquinone***2k** (63%); brown solid; mp: 152–152 °C. IR (KBr) ν_max_ cm^–1^: 3246 (N-H), 3245 (N-H), 1662 (C=O), 1649 (C=O), 1636 (C=O). ^1^H-NMR (CDCl_3_) δ: 3.61 (s, 6H, 2 × OMe), 3.75(s, 3H, OMe), 3.85(s, 3H, OMe), 3.88(s, 3H, OMe), 3.89 (s, 6H, 2 × OMe), 5.99 (s, 2H, H-arom.), 6.13 (s, 2H, quinone + OH), 6.50 (s, 2H, H-arom.), 6.81 (d, 1H, *J* = 8.2 Hz, H-arom.), 6.95 (d, 1H, *J* = 1.8 Hz, arom), 7.25 (dd, 1H, *J* = 8.2, 1.9 Hz, arom), 8.15 (s, 1H, 5-N**H**Ph), 8.34 (br s, 1H, 2-N**H**Ph). ^13^C-NMR (CDCl_3_) δ: 55.9 (2C), 56.2, 56.5 (2C), 61.0, 61.2, 95.8, 101.0 (2C), 103.1 (2C), 107.3, 109.3, 113.2, 125.2, 131.0, 132.4, 132.6, 136.7, 145.7, 146.8, 147.1, 150.6 (2C), 153.1 (2C), 154.1 (2C), 178.2, 179.2, 191.6. HRMS (APCI): [M + H]^+^ calcd. for C_32_H_32_N_2_O_11_: 620.20061; found 620.20726.

*3-(Furan-2-carbonyl)-2,5-bis-[(3,4,5-trimethoxyphenyl)amino]-1,4-benzoquinone***2l** (67%); red solid; mp: 202–203 °C. ^1^H-NMR (CDCl_3_) δ: 3.66 (s, 6H, 2 × OMe), 3.78(s, 3H, OMe), 3.87 (s, 3H, OMe), 3.88 (s, 6H, 2xOMe), 6.12 (s, 1H, quinone), 6.17 (s, 2H, H-arom.), 6.44 (dd, 1H, *J* = 3.6, 1.6 Hz, H-arom.), 6.49 (s, 2H, H-arom.), 6.88 (dd, 1H, *J* = 3.6, 0.8 Hz, H-arom.), 7.47 (d, 1H, *J* = 1.6, 0.8 Hz, H-arom.), 8.11 (s, 1H, 5-N**H**Ph), 8.47 (s, 1H, 2-N**H**Ph). ^13^C-NMR (CDCl_3_) δ: 56.0 (2C), 56.4 (2C), 61.0, 61.1, 96.0, 100.9 (2C), 102.2 (2C), 106.7, 112.9, 117.4, 132.4, 132.5, 136.7, 137.1, 145.7, 146.1, 147.1, 153.4 (2C), 153.5, 154.1 (2C), 177.8, 178.9, 180.2. HRMS (APCI): [M + H]^+^ calcd. for C_29_H_28_N_2_O_10_: 564.17440; found 564.18120.

*3-(Thiophen-2-carbonyl)-2,5-bis-[(3,4,5-trimethoxyphenyl)amino]-1,4-benzoquinone***2m** (64%); red solid; mp: 174.5–175.5 °C. ^1^H-NMR (CDCl_3_) δ: 3.62 (s, 6H, 2 × OMe), 3.79 (s, 3H, OMe), 3.87 (s, 3H, OMe), 3.88 (s, 6H, 2 × OMe), 6.09 (s, 2H, H-arom.), 6.12 (s, 1H, quinone), 6.49(s, 2H, H-arom.), 7.0 (dd, 1H, *J* = 4.9, 3.8 Hz, H-arom.), 7.43 (dd, 1H, *J* = 3.8, 1.2 Hz, H-arom.), 7.55 (dd, 1H, *J* = 4.9, 1.2 Hz, H-arom.), 8.15 (s, 1H, 5-N**H**Ph), 8.44 (s, 1H, 2-N**H**Ph). ^13^C-NMR (CDCl_3_) δ: 56.0 (2C), 56.4 (2C), 61.0, 61.2, 95.9, 101.0 (2C), 103.0 (2C), 107.5, 127.6, 132.2, 132.5, 133.2, 134.6, 136.7, 137.1, 145.6, 145.7, 147.0, 153.3 (2C), 154.1 (2C), 177.8, 178.9, 184.5. HRMS (APCI): [M + H]^+^ calcd. for C_29_H_28_N_2_O_9_S: 580.15155; found 580.15880.

*3-Benzoyl-2,5-bis-[(4-fluorophenyl)amino]-1,4-benzoquinone***2n** (47%); brown solid; mp: 250–251 °C. ^1^H-NMR (CDCl_3_) δ: 6.02 (s, 1H, quinone), 6.68 (t, 2H, *J* = 8.5 Hz, H-arom.), 6.79 (dd, 2H, *J* = 8.7, 4.8 Hz, H-arom.), 7.14 (m, 2H, H-arom.), 7.24 (m, 2H, H-arom.), 7.33 (t, 2H, *J* = 7.6 Hz, H-arom.), 7.48 (t, 1H, *J* = 7.4 Hz, H-arom.), 7.54 (d, 1H, *J* = 1.4 Hz, H-arom.), 7.57 (m, 1H, H-arom.), 8.06 (s, 1H, 5-N**H**Ph), 8.41 (br s, 1H, 2-N**H**Ph). ^13^C-NMR (CDCl_3_) δ: 95.7, 107.5, 115.8, 116.1, 116.7 (2C), 117.1, 125.3, 125.4, 127.7, 127.8, 128.5 (2C), 128.8 (2C), 132.6, 132.7, 132.8, 132.9, 133.4, 137.6, 147.1, 178.4, 179.2, 193.1. HRMS (APCI): [M + H]^+^ calcd. for C_25_H_16_F_2_N_2_O_3_: 430.11290; found 430.11942.

*3-(4-Methoxybenzoyl)-2,5-bis-[(4-fluorophenyl)amino]-1,4-benzoquinone***2ñ** (62%); brown solid; mp: 239–240 °C. ^1^H-NMR (CDCl_3_) δ: 3.86 (s, 3H, OMe), 6.00 (s, 1H, quinone), 6.70 (t, 2H, *J* = 8.5 Hz, H-arom.), 6.81 (dd, 4H, *J* = 8.8, 4.0 Hz, H-arom.), 7.13 (t, 2H, *J* = 8.5 Hz, H-arom.), 7.23 (m, 2H, H-arom.), 7.53 (d, 2H, *J* = 8.9 Hz, H-arom.), 8.09 (s, 1H, 5-N**H**Ph), 8.33 (br s, 1H, 2-N**H**Ph). ^13^C-RMN (CDCl_3_) δ: 55.5, 95.5, 107.7, 113.5 (3C), 115.5, 116.6, 116.9, 125.2, 125.3, 127.6, 127.7, 130.8, 131.2 (2C), 132.4, 132.5, 132.7, 132.7, 145.8, 147.0, 163.7, 178.3, 179.2, 191.4. HRMS (APCI): [M + H]^+^ calcd. for C_26_H_18_F_2_N_2_O_4_: 460.12346; found 460.13003.

*3-(4-Hydroxy-3-methoxybenzoyl)-2,5-bis-[(4-fluorophenyl)amino]-1,4-benzoquinone***2o** (52%); brown solid; mp: 221–222 °C. ^1^H-NMR (CDCl_3_) δ: 3.85 (s, 3H, OMe), 6.01 (s, 1H, OH), 6.11 (s, 1H, quinone), 6.79 (m, 5H, H-arom.), 6.95 (m, 1H, H-arom.), 7.15 (m, 2H, H-arom.), 7.26 (m, 4H, H-arom.), 8.09 (s, 1H, 5-N**H**Ph), 8.28 (s, 1H, 2-N**H**Ph). ^13^C-NMR (CDCl_3_) δ: 56.1, 95.6, 107.8, 109.3, 113.6, 115.6, 115.9, 116.7, 117.0, 125.3, 125.4, 125.5, 128.1, 128.2, 130.9, 132.5, 132.6, 132.7, 132.8, 145.9, 146.8, 147.1, 150.8, 178.4, 179.3, 191.6. HRMS (APCI): [M + H]^+^ calcd. for C_26_H_18_F_2_N_2_O_5_: 476.11838; found 476.12468.

*3-(Furan-2-carbonyl)-2,5-bis-[(4-fluorophenyl)amino]-1,4-benzoquinone***2p** (49%); brown solid; mp: 271.5–272.5 °C. ^1^H-NMR (DMSO-*d*_6_) δ: 5.78(s, 1H, quinone), 6.59 (m, 1H, arom.), 6.90 (m, 4H, H-arom.), 7.05 (d, 1H, *J* = 3.4 Hz, H-arom.), 7.30 (t, 2H, *J* = 8.8 Hz, H-arom.), 7.43 (dd, 4H, *J* = 9.0, 5.0 Hz, H-arom.), 7.85 (m, 1H, H-arom.), 9.50(s, 1H, 5-N**H**Ph), 9.67 (s, 1H, 2-N**H**Ph). ^13^C-NMR (DMSO-*d*_6_) δ: 95.1, 107.2, 112.6, 114.5, 115.0, 116.0, 116.3, 129.2, 126.3 (2C), 127.7, 127.9, 133.9 (2C), 134.0, 134.1, 146.6, 147.1, 147.7, 153.0, 178.1, 178.9, 180.1. HRMS (APCI): [M + H]^+^ calcd. for C_23_H_14_F_2_N_2_O_4_: 420.09216; found 420.09920.

*3-(Thiophen-2-carbonyl)-2,5-bis-[(4-fluorophenyl)amino]-1,4-benzoquinone***2q** (55%); brown solid; mp: 278–279 °C. ^1^H-NMR (CDCl_3_) δ: 5.78(s, 1H, quinone), 6.86 (m, 4H, H-arom.), 7.10 (s, 1H, H-arom.), 7.36 (m, 4H, H-arom.), 7.60 (s, 1H, H-arom.), 7.89 (s, 1H, H-arom.), 9.48 (s, 1H, 5-N**H**Ph), 9.63 (s, 1H, 2-N**H**Ph). ^13^C-NMR (CDCl_3_) δ: 95.1, 107.7, 114.7, 115.0, 116.0, 116.3, 126.2, 126.3, 128.1 (2C), 128.2, 133.7, 133.8, 133.9, 134.0, 134.3, 135.2, 145.1, 146.3, 147.7, 178.1, 178.9, 185.0. HRMS (APCI): [M + H]^+^ calcd. for C_23_H_14_F_2_N_2_O_3_S: 436.06932; found 436.07638.

### 3.3. Biological Assays

#### 3.3.1. Cell Lines and Cell Cultures

Human cancer cell lines T24 (bladder) and DU-145 (prostate), and non-tumor HEK-293 cells were obtained from the American Type Culture Collection (ATCC, Manassas, VA, USA). The cultures were maintained at a density of 1× 10^5^ cells/mL and the medium was changed at 48- to 72-h intervals. They were cultured in high-glucose Dulbecco’s modified Eagle medium (Gibco, Grand Island, NY, USA) supplemented with 10% fetal calf serum, penicillin (100 U/mL), and streptomycin (100 μg/mL). All cultures were kept at 37 °C in 95% air/5% CO_2_ at 100% humidity. Phosphate-buffered saline (PBS) was purchased from Gibco. Cells were incubated at the indicated times at 37 °C, with or without quinones at various concentrations. 

#### 3.3.2. Cytotoxic Assays

The cytotoxicity of the quinones was assessed by following the reduction of MTT (3-(4,5-Dimethylthiazol-2-yl)-2,5-diphenyltetrazolium bromide) to formazan blue [31]. Cells were seeded into 96-well plates at a density of 10,000 cells/well for 24 h and then incubated for 48 h, with or without the quinone derivatives. Doxorubicin was used as the standard chemotherapeutic agent (positive control). Cells were then washed twice with warm PBS and incubated with MTT (0.5 mg/mL) for 2 hours at 37 °C. Blue formazan crystals were solubilized by adding 100 µL DMSO/well, and the optical density of the coloured solutions was subsequently read at 550 nm. Results are expressed as percentage of MTT reduction, compared to untreated control conditions. The IC_50_ values were calculated using the GraphPad Prism software (San Diego, CA, USA).

#### 3.3.3. Quantitative real-time PCR (qPCR) Assay

The T24 cells were cultured as previously mentioned. They were seeded into 6-well plates (2 × 10^5^ cells/well) and, after 24 h of incubation, they were treated for 48 h with **2c** and **2d** (at 32 and 68 µM, respectively). Afterwards, they were washed with phosphate-buffered saline. The cellular lysate was prepared with E.Z.N.A.® RNA-Lock Reagent (Omega Bio-tek, Norcross, GA, USA) to preserve and immediately stabilize the total RNA for the subsequent gene expression assays. The total RNA isolated from the cells using the E.Z.N.A.® HP Total RNA Isolation Kit (Omega Bio-tek) was reverse-transcribed to cDNA using the AffinityScript QPCR cDNA Synthesis Kit (Agilent Technologies, Santa Clara, CA, USA) and 1000 ng of the RNA sample. 

The cDNA synthesized was employed for qPCR using Brilliant III Ultra-Fast SYBR® Green QPCR Master Mix (Agilent Technologies) in a Mx3000P qPCR System (Agilent Technologies), employing a 96-well plate with 20 μL of PCR reaction per well and 10 pmol each of forward and reverse gene-specific primers. Ten genes were analyzed (see Table 4). The relative gene expressions were determined using Beta-2-microglobulin (B2M) as housekeeping, and the delta-delta Ct method (2^−ΔΔCt^ method) with regard to the vehicle-treated group (i.e., the reference group). Five biological replicates were used from each group (treated and reference group). The qPCR reactions were run by duplicates and negative controls contained no cDNA, as previously reported [32,33]. The GraphPad Prism software was used for statistical analyses of the relative gene expressions. The comparisons between means were performed using one-way analysis of variance (ANOVA) and Dunnett’s multiple comparisons test. All statistical analyses were performed with a significance level of *p* < 0.05. 

## 4. Conclusions

In summary, we have synthesized of a number 3-acyl-2,5-*bis*(phenylamino)-1,4-benzoquinones and assessed their voltammetric values and cytotoxicities of cancer cells in vitro. The members of the series **2a**–**q** were prepared from acylbenzoquinones and phenylamines, in moderate to good yields (47–74%). The first and second half-wave potential values (E _1/2_) of the members of the series were sensitive to the push-pull electronic effects of the substituents around the benzoquinone nucleus, as shown by the cyclic voltammograms of the aminoquinones. The preliminary results of the biological evaluation of the compounds **2a**–**q** showed interesting in vitro cytotoxic activity on cancer cells. In this context, there were two more active compounds which increased TNF gene expression, suggesting an association with an inflammatory-like response.

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
