# Peer review of "Half-Wave Potentials and In Vitro Cytotoxic Evaluation of 3-Acylated 2,5-*Bis*(phenylamino)-1,4-benzoquinones on Cancer Cells"

_molecules, 2019, doi:10.3390/molecules24091780_

Round 1

Reviewer 1 Report

Comments for Manuscript Molecules-484885

Manuscript Molecules-484885 reported the syntheses of a series of -acyl-2,5-bis(phenylamino)-1,4-benzoquinones, along with their voltammetric values and in vitro cancer cell cytotoxicity. Among them, compounds 2c and 2d showed the highest cytotoxicity with good selectivity. Thus, this manuscript is recommended to publish on Molecules after major revision.

Major concerns:

1. The authors reported the voltammetric values (E1/2) of these compounds, and is there any connection with their cytotoxicity?

2.  These compounds are reported with redox properties. It seemingly will affect the MTT evaluation method, which is based on the NAD(P)H-dependent cellular oxidoreductase enzymes.

Minor concerns:  

1. The authors reported a novel synthetic method to yield 2a-q, which will be interesting to propose a plausible mechanism for this step.

2. The structures of the products were determined by 1D-NMR and 2D-NMR, so the HMBC spectrum should be attached. 

Author Response

Reviewer 1

Manuscript Molecules-484885 reported the syntheses of a series of 3-acyl-2,5-bis(phenylamino)-1,4-benzoquinones, along with their voltammetric values and in vitro cancer cell cytotoxicity. Among them, compounds 2c and 2d showed the highest cytotoxicity with good selectivity. Thus, this manuscript is recommended to publish on Molecules after major revision.

Answer: we thanks the referee for her/his valuable and kind recommendation.

Major concerns:

1.      The authors reported the voltammetric values (E1/2) of these compounds, and is there any connection with their cytotoxicity?

Answer: according to the obtained data, cytotoxicity did not appear to be associated with redox potential values. Conversely, it seems that the second molecular descriptor, namely ClogP, would have a potential role. Indeed, the most active compounds have high values of ClogP (2.05 and 2.88, respectively. However, other derivatives with similar values of lipophilicity (2n, 2ñ and 2q) were devoid of cytotoxicity. We concluded that it is rather difficult to associate a given biological response with only one molecular descriptor.

2.      These compounds are reported with redox properties. It seemingly will affect the MTT evaluation method, which is based on the NAD(P)H-dependent cellular oxidoreductase enzymes.

Answer: Such a situation is rather unlikely. First, the experimental procedure involves a two-times cellular washing after the 48h incubation in the presence of compounds. Consequently, the few amount of resting molecules will not interfere with the MTT reduction. Secondly, even in a very unlikely situation in which such few amounts would affect the enzyme reaction, then similar results would been obtained for most of quinones. This was not the case.

Minor concerns:  

1.      The authors reported a novel synthetic method to yield 2a-q, which will be interesting to propose a plausible mechanism for this step.

Answer: In fact, the reaction of some 2-acetyl-1,4-benzoquinone with phenylamines to produce 3-acetyl-2,5-bis(phenylamino)-1,4-benzoquinones is reported in literature (please see refs 13). In our paper we report a new procedure and the scope has been extended to a broad number of analogs.

2.      The structures of the products were determined by 1D-NMR and 2D-NMR, so the HMBC spectrum should be attached. 

Answer: Examples of NMR spectra of compounds 2b,2g,2h,2j,2m,2ñ,2p were attached as Supplementary Materials

Reviewer 2 Report

I suggest the following modifications before this paper is suitable for publication.

(1) The NMR spectra (including 1H, 13C, and HMBC) for 2a-q should be provided as Supporting Information and need to be carefully examined.

(2) The method to obtain minimal energy conformation of 2b and 2p need to be described.

Author Response

Rewiewer 2

We thanks the referee for her/his valuable and kind recommendation.

(1)   The NMR spectra (including 1H, 13C, and HMBC) for 2a-q should be provided as Supporting Information and need to be carefully examined.

Answer: Examples of NMR spectra of compounds 2b,2g,2h,2j,2m,2ñ,2p were attached as Supplementary Material.

(2) The method to obtain minimal energy conformation of 2b and 2p need to be described.

Answer: The method concerning to the molecular models in figure 1 was included in the following paragraph:

It is worth mentioning that inspection of minimal energy conformation of compounds 2b and 2p, performed by MM2 calculation (ChemBio3D 11.0), shows coplanar orientation between the benzoquinone nucleus and the phenylamino substituent in C-5. Furthermore, it was also observed that rotation of the substituents linked to the 2,3-quinone double bond are strongly hindered.

Round 2

Reviewer 1 Report

The concerns have been well addressed. This manuscript is suggested to be published on Molecules.